# Time-resolved single dopant charge dynamics in silicon

Mohammad Rashidi[1,2], Jacob A.J. Burgess[3,4], Marco Taucer[1,2], Roshan Achal[1], Jason L. Pitters[2], Sebastian Loth[3,4] & Robert A. Wolkow[1,2]

As the ultimate miniaturization of semiconductor devices approaches, it is imperative that the effects of single dopants be clarified. Beyond providing insight into functions and limitations of conventional devices, such information enables identification of new device concepts. Investigating single dopants requires sub-nanometre spatial resolution, making scanning tunnelling microscopy an ideal tool. However, dopant dynamics involve processes occurring at nanosecond timescales, posing a significant challenge to experiment. Here we use time-resolved scanning tunnelling microscopy and spectroscopy to probe and study transport through a dangling bond on silicon before the system relaxes or adjusts to accommodate an applied electric field. Atomically resolved, electronic pump-probe scanning tunnelling microscopy permits unprecedented, quantitative measurement of time-resolved single dopant ionization dynamics. Tunnelling through the surface dangling bond makes measurement of a signal that would otherwise be too weak to detect feasible. Distinct ionization and neutralization rates of a single dopant are measured and the physical process controlling those are identified.

[1] Department of Physics, University of Alberta, Edmonton, Alberta, Canada T6G 2J1. [2] National Institute for Nanotechnology, National Research Council of Canada, Edmonton, Alberta, Canada T6G 2M9. [3] Max Planck Institute for the Structure and Dynamics of Matter, 22761 Hamburg, Germany. [4] Max Planck Institute for Solid State Research, 70569 Stuttgart, Germany. Correspondence and requests for materials should be addressed to M.R. (email: rashidi@ualberta.net).

Understanding the fundamental properties of individual dopants locally coupled to electronic transport in semiconductors is the focus of much current research[1]. This area of enquiry holds promise for new semiconductor device applications at the ultimate level of miniaturization, as well as for utilization of the quantum properties of individual impurities[2]. Scanning tunnelling microscopy (STM) has enabled significant progress in this field, including observation of the influence of single dopants on the local density of states[3–6] and local magnetic properties[7,8] of the host semiconductor. Measurements on GaAs have shown slow dopant dynamics can be observed locally with millisecond timescale STM[9]. Dopant dynamics must exist on much faster time scales as well, but have not yet been observed using STM. Time-resolved STM (TR-STM), pioneered by Nunes and Freeman[10], has recently seen a resurgence of interest[11–15] following the application of a simplified all-electronic technique to the measurement of spin dynamics in adatoms[16]. Thus far, this method has only been applied to other magnetic adatom systems[12] and atomically assembled nanomagnets[13]. Nanoscale dynamics of optically excited charge carriers on semiconductor surfaces[17] and quantum dots[18] were observed with optical pump-probe STM techniques. However, the purely electronically excited dynamics of dopants have not been studied with high time-resolution using STM.

Here we measure single electron charge dynamics of individual subsurface dopants with time-resolution well beyond the limitations of conventional STM. We perform point measurements, drawing current through a surface dangling bond (DB) on a hydrogen terminated arsenic-doped Si(100)-2 × 1 sample with a dopant depletion layer at its surface (Fig. 1). Dopants and the DB are decoupled from the bulk by this layer. Unlike the hydrogen terminated regions of the surface, which are passivated, the spatially extended DB orbital provides strong overlap to both the tip orbital and, via vibronic coupling, to the bulk bands[19], greatly enhancing sensitivity to single dopant effects. Ionization of a dopant in the electric field of the tip opens a conductance channel from the conduction band to the tip via the surface DB. This allows detection of the dopant charge state in a method analogous to a single atom gated transistor: the bulk acts as the source, the DB (strongly coupled to the tip) is the drain, and the dopant is the gate. A combination of fast real-time acquisition and pump-probe techniques enables temporal mapping of the local dopant dynamics from nanoseconds to seconds by exploiting the DB's amplification of the single dopant effects.

## Results

**Conventional STM and spectroscopy.** Scanning tunnelling spectra ($I(V)$ measurements) of DBs on samples heated to 1,250 °C during oxide desorption are correlated with a sharp current onset (Fig. 2a) at a critical filled-state bias voltage[20]. At sample biases less negative than the critical voltage, the DB appears dark with a spatially extended halo (right panel in Fig. 2b). Past the critical voltage, the DB appears bright (left panel in Fig. 2b). At the critical voltage (middle panel in Fig. 2b), some DBs appear striped and exhibit stochastic two-state switching between low-conductance and high-conductance states at millisecond timescales (Fig. 2c,d). The sharp step in $I(V)$ measurements and the associated current fluctuations at the critical voltage have not been explained previously, but it has been argued that they can be related to a change in the supply of electrons to the DB from the bulk silicon[20].

Secondary ion mass spectroscopy (SIMS) shows that as a result of flashing to 1,250 °C the sample is depleted of dopants in the near surface region (up to depth of ∼60 nm) (ref. 21). The black curve in Fig. 3a shows the conduction band edge calculated near

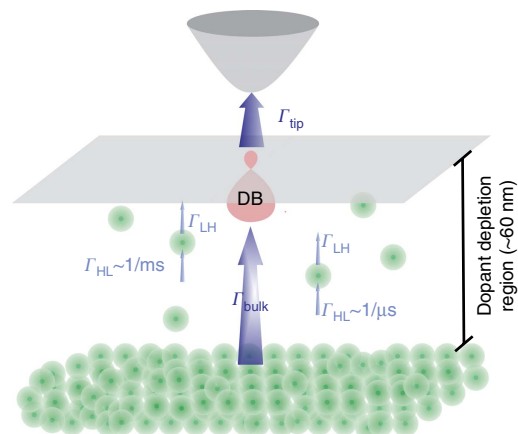

**Figure 1 | Schematic of the system of study presenting the dopant concentration in the vicinity of a dangling bond.** Arsenic dopants are indicated by green balls. The relevant rates probed in our measurements are indicated by arrows. $\Gamma_{HL}$ and $\Gamma_{LH}$ denote the dopants' electron filling rate from the bulk and their emptying rate by the tip field, respectively. $\Gamma_{bulk}$ and $\Gamma_{tip}$ denote the DB's electron filling rate from the bulk and its emptying rate by the tip, respectively.

the surface considering the band gap narrowing[22], tip-induced band bending (TIBB)[23] and the dopant concentration profile from SIMS measurements[20,21]. As a result of the dopant depletion region, the degenerately doped bulk does not extend to the surface. The conduction band edge first rises as a result of the depletion of dopants in that region, and falls rapidly as a result of TIBB at the surface, acting as an energy barrier for the electrons to conduct from the bulk (degenerately doped region) to the surface. This potential barrier accounts for the weak supply of electrons from the bulk to the DB for sample biases less negative than the critical voltage. As the critical voltage is crossed, there is a sudden change in the supply rate of the DB. Similar features in other systems have been attributed to the ionization of dopants in the field of the tip[3–6]. The electrostatic potential due to an ionized dopant significantly reduces this energy barrier (green curve in Fig. 3a), leading to an increase in the electron conductivity from the bulk to the surface. This understanding of the sharp onset in $I(V)$ measurements also provides a potential explanation for the related current fluctuations shown in Fig. 2c. Tunnelling current fluctuates, exhibiting two state noise, suggesting it could be due to an individual dopant switching charge state between neutral and positive. The magnitude of the influence of a nearby dopant ionization can be estimated by computing transmission through the energy barrier, using the Wentzel–Kramers–Brillouin approximation (Supplementary Fig. 1). Additionally, the population of thermally excited electrons in the conduction band edge rises significantly for barriers of <5 meV at our experimental temperature (4.5 K). In conjunction with ionizing dopants, either tunnelling or thermal excitation can explain the electron transport from the bulk to the surface through the energy barrier and the observed sudden current turn on in $I(V)$ spectroscopy.

Our model of dopant ionization allows us to investigate the nature of the dopants causing the gating. Subsurface dopants within the first few atomic layers which can be imaged using STM[5,6,24] are already ionized because of strong TIBB in our experiment. Moreover, calculations using a Wentzel–Kramers–Brillouin approximation to compute transmission from the bulk to the DB at the surface (Supplementary Fig. 1) show that near surface dopants have a small gating effect. We, therefore, cannot correlate the gating effect with any nearby shallow subsurface

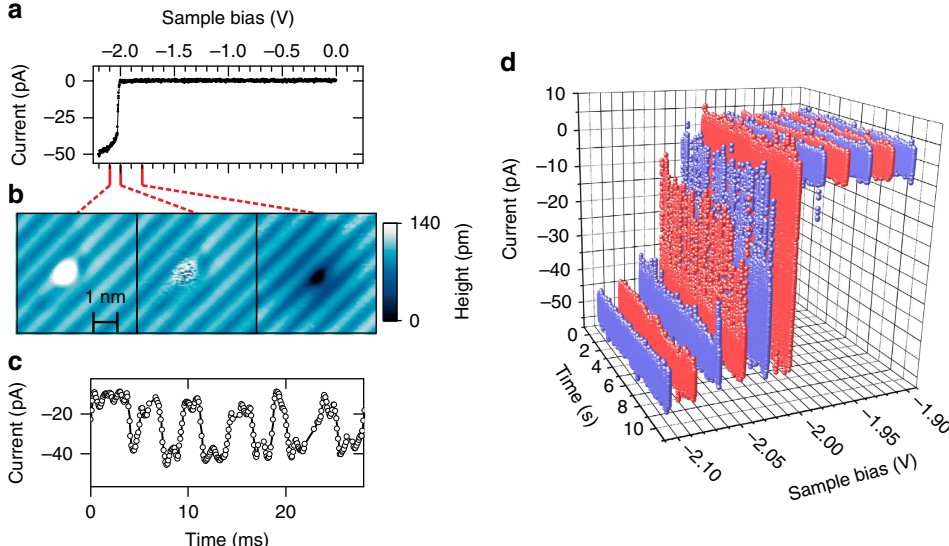

**Figure 2 | Spectroscopic behaviour of selected dangling bonds on Si(100) samples with dopant depletion layer at the surface.** (**a**) Constant-height filled-state $I(V)$ spectrum measured over a DB. (**b**) Constant-current STM images of the DB beyond ($-2.1$ V), at ($-2.0$ V) and before the critical voltage range ($-1.8$ V) at left, right and centre, respectively. The silicon dimer rows of the Si(100) $- 2 \times 1$ surface show up as bars in the STM topographs. (**c**) Current time-trace at the critical voltage ($-2.01$ V) exhibiting telegraph noise behaviour at millisecond timescales. (**d**) Current-time traces in the critical bias range acquired for 11 bias voltages from $-1.90$ to $-2.10$ V with the maximum sampling rate of the STM controller ($\sim 10$ kHz). Alternating colours are used for better visualization.

dopants that can be imaged with standard STM techniques. We estimate that at the critical bias voltage only the dopants within $\sim 15$ nm of the surface can be ionized by the tip field. Therefore, only dopants that are located between 5 and 15 nm below the surface within a 15 nm lateral region are estimated to have a major role in opening the conduction channel. Based on the dopant profile measured by SIMS[20,21], there exist on average $<10$ dopants in this volume. Since these dopants are randomly distributed a large variety of switching behaviours are expected and indeed observed for different DBs.

Alternative explanations of the sharp step in $I(V)$ spectra and of the current fluctuations involving transitions of the charge state of the DB itself need to be examined. The DB has three allowed charge states, positive, neutral and negative. These charge states are associated with two charge transition levels from positive to neutral and from neutral to negative, denoted by $(+/0)$ and $(0/-)$, respectively. A sharp step in $I(V)$ can often be attributed to the tip Fermi level coming into resonance with a charge transition level[25]. However, we are able to rule out such an explanation here because the tip Fermi level at the critical bias regime is already significantly below both charge transition levels of the DB[20] (Fig. 3b).

DB charge state fluctuation itself can also be ruled out as a possible source of the slow dynamics (Fig. 2c,d). The current time traces that show the telegraph noise are recorded with the tip placed exactly on top of the DB, with the initial tip height set at a voltage where the DB shows a bright appearance. Therefore, the tip is removed from the surface by an additional $\sim 200$ pm compared with the hydrogen-terminated silicon surface. This effectively eliminates electrons tunnelling directly from the silicon valence band. The only sufficiently conductive tunnelling pathway that remains is the one that originates from the conduction band and passes through the DB. Since the tunnel current passes through the DB, the fluctuations in the DB charge state must occur at timescales less than $e/I_T = 2$ ns to 20 ns; much faster than the switching observed. This contrasts with a recent STM study[26] that considered millisecond dynamics of the DB caused by

changes in its charge state. Crucially, to detect this effect the STM tip had to be placed 2 to 4 nm away from the DB, laterally. As a result, the current which flowed through the DB was negligible ($\sim 10^{-6}$ pA to $10^{-4}$ pA, far below our sensitivity), and the much larger direct tip-sample tunnelling current was gated by the electrostatic potential of the charged DB. In this work, since the STM tip is placed at the DB, and for the reasons outlined above, we can rule out this type of explanation. We are, therefore, compelled to look for a change in the DB's environment, which alters, or gates, the dominant current pathway. This provides further evidence that the fluctuating charge state of a dopant near the DB modifies the supply of current from the bulk to the DB. With this understanding of the spectroscopy and of the observed current fluctuations, we now turn our attention to measurements of time-dynamics spanning the range from milliseconds to nanoseconds, and compare these to a model of single dopant ionization.

The supply and drain rate of any dopants ($\Gamma_{HL}$ and $\Gamma_{LH}$ shown schematically in Fig. 1) are also subject to electrostatic effects, and the dynamics should, therefore, show a dependence on bias voltage. This may be probed in a limited way by measuring telegraph noise traces acquired as a function of voltage (Fig. 3c). We find that the transition rate from the high-conductance to the low-conductance state exhibits a minimal modification with bias, while the opposing transition $\Gamma_{LH}$ speeds up drastically with more negative bias voltage. At $-2.02$ V it reaches the bandwidth limit of our current amplifier making it a challenge to substantiate the model of a dopant acting as a gate. In order to extend our measurements to the nanosecond time scale, we employ all-electronic pump-probe STM.

**Time-resolved STM and spectroscopy.** We apply three variations of pump-probe STM: a pulsed form of $I(V)$ spectroscopy used to search efficiently for fast dynamics, a typical pump-probe experiment to measure the relaxation dynamics of the high-conductivity state, and a variable pump width experiment

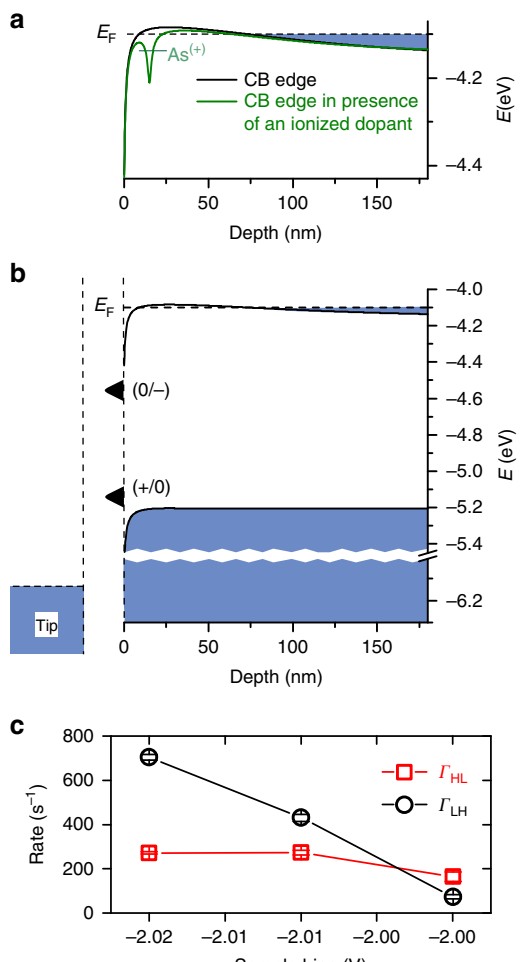

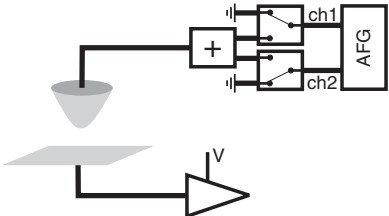

**Figure 4 | Time-resolved scanning tunnelling microscopy set-up.** An arbitrary function generator is the source of the pump and probe pulse trains, which are applied to the STM tip. Two radio frequency switches are used to turn on and off either pulse train. The DC bias voltage is applied to the sample and the tunnel current is measured at the sample side as well.

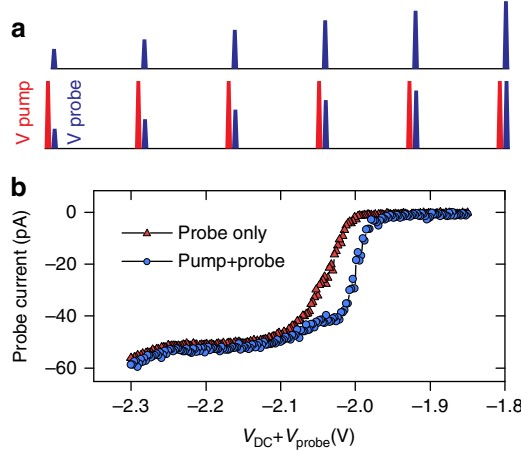

**Figure 5 | Time-resolved scanning tunnelling *I(V)* spectroscopy (TR-STS).** (**a**) Schematic of TR-STS pulse timing: tunnelling current is measured from a series of varying-amplitude probe-pulses (blue) with and without a preceding pump pulse (red). (**b**) TR-STS with 1 μs probe pulses with (blue circles) and without (red triangles) a preceding 1 μs pump pulse. The DC bias voltage is set at −1.80 V. The amplitude of the pump pulse is 0.50 V and that of the probe pulse is swept from 50 to 500 mV. The relative delay between the trailing edge of the pump and the leading edge of the probe pulse is 10 ns and the repetition rate is 25 kHz. The observed hysteresis in TR-STS overlaps the bias range over which the system is bistable.

**Figure 3 | Band diagram of the system of study influenced by an ionized dopant and analysis of the slow transitions in the bistable bias range** (**a**) Conduction band edge in the presence (green curve) and absence (black curve) of an ionized dopant computed for the sample bias voltage of −2.0 V. (**b**) Full band diagram of the system of study. (+/0) and (0/−) are the DB's charge transition levels from positive to neutral and neutral to negative, respectively. Blue coloured area indicates filled states. (**c**) Rates extracted from current time-traces in the bistable bias range, calculated by signal pair analysis method (Supplementary Note 1).

to measure the excitation dynamics of the transition from the low-conductance state to the high-conductance state. The experimental scheme for applying pulses to the STM is very simple. Pulses are applied directly to the STM tip, and a pair of radio frequency switches are used to turn off and on the pump and probe pulse trains (Fig. 4). Common to all three types of experiment is the role of the pump and probe. The pump transiently brings the bias level from below to above the critical voltage, driving the transition to the high-conductance state. In the context of a dopant gate, the pump increases TIBB making it more likely for a dopant to ionize and improve electron supply to the DB. The probe follows at a lower voltage to sample the conductance after the pump.

Figure 5a displays a schematic of time-resolved *I(V)* spectroscopy used to search for dynamics. Instead of sweeping the DC bias voltage as in standard *I(V)* spectroscopy, we keep the DC bias fixed and sweep the amplitude of a train of fast microsecond scale voltage pulses. Preceding each probe pulse, a large fixed amplitude pump pulse transiently disturbs the system. The delay

between pump and probe is fixed such that the pulses do not overlap (10 ns between pulse edges). Thus, the transient state immediately resulting from the pump is interrogated by the probe. Since the current is almost zero at the DC offset, the measured tunnelling current from the preamplifier comes exclusively from current during the time when the pulses are on. Comparison to an equivalent curve measured with no pump pulse reveals any triggered dynamics. The DB shown in Fig. 2, exhibits deviations between the two curves starting at the critical voltage, −2.00 V, and extending up to −2.10 V (Fig. 5b). By contrast, the slow telegraph noise is only observed up to −2.05 V, indicating the presence of additional nanosecond scale dynamics.

We investigate this region using a conventional pump-probe measurement, using fixed amplitude pulses and a variable delay between pump and probe. This measurement captures the relaxation dynamics from the DB's high-conductance state to the low-conductance state. As shown schematically in Fig. 6a, to measure the time constant of this transition from high to low conductance ($\tau_{HL}$), the DC offset is set at a sample bias voltage where the system is in the low-conductance state. A pump pulse brings the system to the high-conductance state and a subsequent

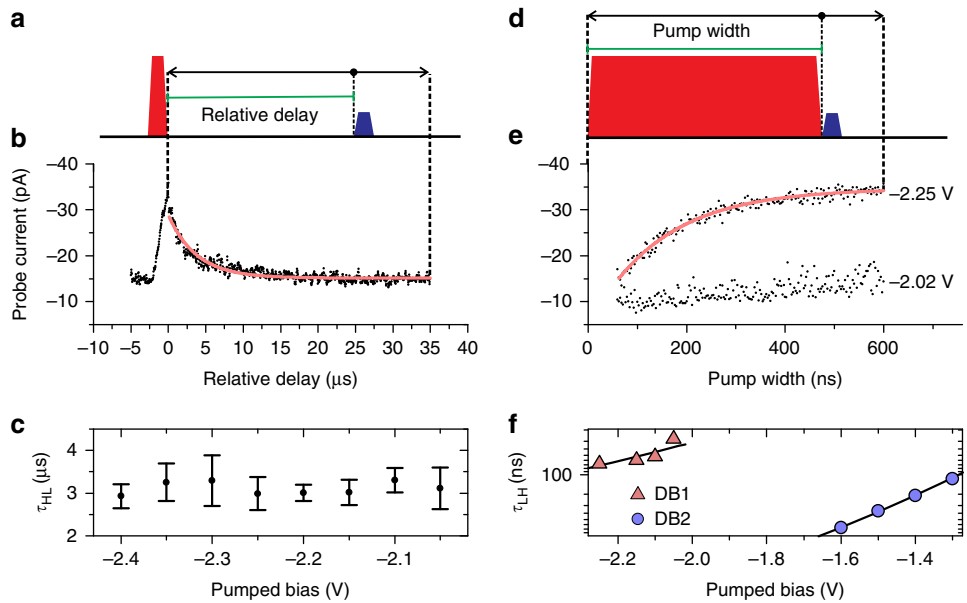

**Figure 6 | Time-resolved STM measurements of relaxation and excitation time constants.** (**a**) Schematic of the relaxation time measurement $\tau_{HL}$. The green bar indicates the relative delay, which is defined as the time between the trailing edge of the pump pulse and the leading edge of the probe pulse. (**b**) $\tau_{HL}$ measurement from a 1 μs width probe pulse and a 1 μs width preceding pump pulse with repetition rate of 25 kHz at different relative delays. The probe (pump) amplitude is −0.21 V (−0.40 V). With the DC offset set at −1.80 V, the probe and pump pulses are at −2.01 and −2.20 V, respectively. $\tau_{HL}$ is extracted from this measurement as the decay constant of a single exponential decay function fitted to the experimental data. (**c**) Measured $\tau_{HL}$ for different pump amplitudes shows that the time constant is independent of the pump bias. Error bars indicate standard errors of exponential fittings. (**d**) Schematic of the excitation time measurement $\tau_{LH}$. The green bar indicates the width of the pump pulse. (**e**) $\tau_{LH}$ measurements at different pumped bias voltages (−2.02 and −2.25 V shown as two examples) using a variable width pump pulse. The DC offset is −1.80 V, pulse amplitudes are the same as in **b**, the probe width is 1 μs and the relative delay is 10 ns. Red curves in **b** and **e** are the single exponential fits. The time constants of the exponential fits correspond to $\tau_{LH}$. (**f**) $\tau_{LH}$ at different pumped bias voltages for the DB in Fig. 1 (denoted here as DB1) and for DB2, which is fully described in the Supplementary Figs 2–4. Solid lines are the fits using equation (1).

probe pulse at different relative delays probes the status of the system at a sample bias at which bistability occurs. Within the relative delay time the applied bias is the DC offset, which has the effect of letting the system decay to the low-conductance state. At zero relative delay, the system is in the high-conductance state. As the relative delay is increased, the system transitions to the low-conductance state. The time constant of the exponential decay is a measure of the characteristic time over which the system randomly transitions from the high-conductance state to the low-conductance state. Specifically, it is a measure of the time it takes for a dopant to be supplied with an electron.

Figure 6b shows the probe pulse current contribution as the pump-probe delay is swept. The DB is excited to a state with higher conductance by the pump, and subsequently relaxes back to the low-conductance state. The relaxation can be fit with a single exponential decay function to extract the time constant of the high-to-low relaxation, $\tau_{HL}$. We do, however, see an offset current; the low-conductance state is not zero. Such a contribution is consistent with the pump pulse simultaneously exciting both the fast dynamics, and slow dynamics seen in telegraph noise. At the repetition rate appropriate for inspecting the fast dynamics, there is insufficient time for the slower process to relax, leading to an approximately constant offset current. We search for additional signs of time scale changes as a function of pump pulse amplitude. Between pump voltages of −2.40 and −2.05 V we find an approximately constant lifetime of $\tau_{HL} = 3.1 \pm 0.1$ μs (Fig. 6c). This indicates that, beyond the change in time scale at −2.05 V we do not find any additional thresholds on this DB, implying that the DB is only affected by two dopants.

To probe the excitation dynamics, we perform a pump-probe experiment varying the pump length and probing at a fixed delay.

This measurement captures the time constant for transition from low- to high-conductance state, $\tau_{LH}$ (Fig. 6d). A pump pulse with varying width is applied to bring the system to the high-conductance state, and a subsequent probe pulse shortly after (10 ns) checks the state of the system at a sample bias at which bistability is observed. In order to calculate the current from the probe pulse only, for each pump width, the signal is measured with a small amplitude probe pulse and the result is subtracted from the measured signal with the probe pulse at the bistable bias range. The DC bias offset is maintained at sample biases less negative than the bistable bias range to allow the system to relax to the low-conductance condition in the intervening time between pump-probe pairs. For short pump width, the system stays in low-conductance state and the probe current is almost zero. Figure 6e shows increasing probe pulse current contribution as the pump pulse widens. The time constant of the exponential increase gives the time needed to excite the system from the low-conductance state to the high-conductance state during the pump, $\tau_{LH}$. In other words, it gives the time needed to ionize a dopant. Unlike $\tau_{HL}$, $\tau_{LH}$ shows a strong variation with changing pump amplitude. The improved bandwidth afforded by pump-probe investigations allow us to identify the dependence as exponential (Fig. 6f).

The results in Fig. 6e,f can be understood in terms of tunnel ionization of a neutral dopant in the field of the STM tip (Supplementary Note 2 and Supplementary Fig. 5). At negative sample bias, dopants in the surface region are below the Fermi level. However, due to the non-equilibrium situation as a result of the depletion of surface dopants, beyond a critical voltage the emptying rate exceeds the supply of electrons from the bulk, causing the ionization of the isolated near-surface dopants.

As arsenic is a hydrogenic dopant in Si, the tunnel ionization time constant can be approximated as[27]:

$$\tau_{LH} = \frac{e^3 F}{128\pi\varepsilon E_{As}^3}\exp\left(\frac{32\pi\varepsilon E_{As}^2}{3e^3 F}\right), \qquad (1)$$

where $E_{As}$ is the binding energy of the arsenic dopant and $\varepsilon$ is the dielectric constant of silicon. The ionization rate depends exponentially on the local electric field strength, $F$, at the dopant. The field strength experienced by a dopant in the depletion layer depends on its local environment and the electric field arising from TIBB[23]. Hence, the tip-induced electric field at any dopant increases proportionally to the applied bias and may be approximated as $F = \kappa|V| + F_0$, where $\kappa$ is a proportionality factor relating the electric field to the applied bias, $V$, and $F_0$ is a constant field offset that includes the contact potential difference between tip and silicon surface and interactions with nearby ionized dopants. This model accurately describes the observed switching rates shown in Fig. 6f: $\tau_{LH}$ depends exponentially on bias for both DBs shown but the rates are offset in bias as expected for different local dopant distributions.

Dopant ionization is exponentially dependent on the applied bias (Fig. 6e and equation (1)), but dopant supply is determined by the local environment and is roughly independent of bias (Fig. 3c) as the tip field extends only $\sim 15$ nm below the surface. Therefore, the onset of current occurs in a narrow transition region near the critical voltage where ionization overtakes dopant supply. We have examined a few hundred DBs by our technique. In most cases, the supply rate of electrons to the dopant was in the Hz to kHz range, and we were able to observe the corresponding dynamics using standard $I(t)$ measurements. In few cases, the supply rate was in the kHz to MHz range, and we have used pump probe to observe these dynamics near the cross-over of these two rates. Variation in DB-dopant separation leads to variation in the opening voltage (observed between $-0.7$ and $-2.0$ V) of this channel in standard $I(V)$ STS measurements among individual DBs[20].

The DB shown in Fig. 2 exhibits dynamics at two distinct time scales. We attribute this to the influence of two isolated dopants with different supply rates one slow, and one fast from the bulk. These two dopants are schematically displayed in Fig. 1. Our results show that there is a distinct voltage threshold for each dopant. Figure 6d,e show that we begin to find fast dynamics at approximately $-2.05$ V, that is, pumping to less negative values than this threshold voltage does not result in any measurable dopant ionization time constant. This is notable in Fig. 5b also as the splitting between the curves remains open for bias voltages more negative than approximately $-2.05$ V where slow telegraph noise is no longer measurable (Figs 2d and 3c). Therefore, we assign approximately $-1.80$ V where the slow dynamics start to appear (Fig. 2d) as the threshold voltage to ionize the slow dopant and approximately $-2.05$ V to the ionization of the fast dopant.

## Discussion

Our model permits a robust description of the dynamics observed and demonstrates that the sharp tunnelling current features stem from the gating influence and ionization thresholds of nearby dopants. More importantly, in conjunction with the ability of TR-STM to identify discrete voltage thresholds triggering separate time scales of dynamics, the model permits the disentanglement of the influence of multiple dopants. Here we analyse a DB affected by two dopants, one slow, and one fast as an example that captures a broad range of behaviour. The expected range of variability in numbers and interactions of nearby dopants readily explains the wide range of critical voltages and sets of multiple current steps (Supplementary Fig. 6) seen in measurements of

other DBs[20]. In addition, the statistical distribution of the dopants together with the extreme dependence of the switching rate on local potential makes it such that we can expect dopant dynamics at any timescale. In this paper, we can capture time dynamics in a large range from seconds to nanoseconds. To achieve this, we have combined different techniques and can find dynamics on all accessible timescales though not always for an individual DB. This is unsurprising because of the limited number of interacting dopants.

The silicon surface dangling bond is a convenient and powerful probe for dynamics of local dopants that may be used with time-resolved scanning tunnelling microscopy to understand complex multi-dopant environments. The combined technique and system has great potential in exploring the effects of sparse dopants in transistors, and in prototyping novel devices based on interacting or isolated dopants.

## Methods

**Experimental parameters.** Measurements were performed using an Omicron low temperature STM and Nanonis control system at a temperature of 4.5 K with a base pressure of $5 \times 10^{-11}$ Torr. The STM is commercially equipped with radio frequency wiring with a bandwidth of $\sim 500$ MHz. The tungsten tips were chemically etched and cleaned and sharpened by electron beam heating followed by field ion microscopy[28]. The samples were cleaved from a 3–4 m$\Omega$ cm n-type arsenic doped Si(100) wafers (Virginia Semiconductor Inc.). They were degassed for several hours at 600 °C and were cleaned by multiple flash annealing to $\sim 1,250$ °C then hydrogen-terminated near 330 °C (ref. 29). The high-temperature flash anneal depletes dopants in a surface region 60 nm wide as measured by SIMS[21]. $I(V)$ spectra were measured in constant-height mode and tip quality was checked by performing reference spectroscopy on bare silicon and bare dimer defects. Voltage pulses and gentle, controlled contacting of tip to sample was used to improve the tip quality for STM imaging and spectroscopy in-situ.

**Time-resolved scanning tunnelling microscope set-up.** As shown schematically in Fig. 4, cycles of voltage pulse pairs were generated by an arbitrary function generator (AFG), Tektronix AFG3252C, summed (Mini-Circuits ZFRSC-42-S + ), and fed into the tip. The DC bias voltage was applied to the sample and the tunnelling current was collected through an Omicron preamplifier, connected to the sample. Two radio frequency switches (Mini-Circuits ZX80-DR230-S + ) connected to AFG output channels were used to either ground the tip during STM imaging and spectroscopy or connect the tip to the AFG during pump-probe measurements. The auto-correlation signal measured on H:Si was used to monitor the quality of the pulses at the tunnel junction. Ringing is a common problem in this technique due to the imperfect impedance match between the tunnel junction and 50 $\Omega$ impedance of the rest of the set-up[11]. To mitigate the ringing, 25 ns wide pulse edges were used throughout this work.

For time-resolved measurements, tunnelling current is induced by a series of short voltage pulses rather than by continuous DC bias. A significant number of parameters can vary for the measurement. Pulse amplitude, pulse width, repetition rate, delay between pump/probe pulse pairs and total measuring time can be adjusted depending on the nature of the measurement.

**Data availability.** The data that support the findings of this study are available from the corresponding author upon request.

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

## Acknowledgements

We would like to thank Martin Cloutier and Mark Salomons for their technical expertise. We also thank NRC, NSERC, and AITF for financial support. J.A.J.B. acknowledges postdoctoral support from NSERC and the Alexander von Humboldt Foundation.

## Author contributions

M.R., J.A.J.B. and M.T. designed and performed the experiments, analysed the data and co-wrote the paper. R.A. did the signal pair analysis. J.L.P. contributed to the interpretation of the results. S.L. and R.A.W. supervised the project. All authors discussed the results and commented on the manuscript.

## Additional information

**Competing financial interests:** The authors declare no competing financial interests.

