## [Peer review file · Nature Communications]

Reviewers' comments:

Reviewer #1 (Remarks to the Author):

The authors have adequately addressed the previous concerns of the referees. The revised version of the manuscript is suitable for publication in Nature Communications.

Reviewer #2 (Remarks to the Author):

The manuscript by Rashidi et al., reports the time-resolved STM studies of individual dangling bonds in a Silicon surface. This is the first study of dopant dynamics in a semiconductor using a novel time resolved technique pioneered by co-author Loth for studies of magnetic adatoms. In my opinion, I believe the extension of this technique to study dynamics of individual dopants in a semiconductor is an important result that would be of broad interest to researchers in the nanoscience and semiconductor research communities. However, I don't think the manuscript is clear enough to make the physics understandable to non experts, and the omission of measurement details and organization would make it very difficult to understand in a reasonable amount of time for experts in the field. In my opinion the manuscript needs some structural reorganization and more significant discussion of points I list below to possibly be suitable for publication in an NPG journal.

1) I think the authors could make more explicit references to their earlier work to make it clear that they have some idea of the concentration of the dopants in the depletion region (I.e. include a number even if only an order of magnitude), and I would feature the calculations in Fig. S7 more prominently in the main text. Given their concentration estimates and this calculation in fact, one can estimate that it's reasonable for only a handful of As dopants to influence the DB tunneling, and that the subsurface As which they can image with STM wouldn't be the primary contributors. According to this estimate, it would be dopants $\sim 30\text{nm}$ directly below the tip which would have the biggest effect; if this is true, I'd revise line 148 which estimates 'several nm'.

2) The text seems to be a bit unclear on whether there are two or three DB charge states (e.g., +1, 0, -1) accessible in the STM experiments. This is left ambiguous in line 63 in the text, but seems to be better discussed in the groups' recent PRL and New J Phys publications. A much more complete schematic of the expected tunneling channels and relevant rates would be helpful in this manuscript, more along the lines of what the group has done in their recent publications. This would help the expert appreciate why the switching behavior reported here may be qualitatively different than very similar behaviors observed under different tunneling conditions in the earlier work.

3) A key point that lacks discussion in the manuscript is some rationale for why they see two such disparate timescales for dynamics. The authors suggest this is due to the influence of two dopants (inclusion of two As dopants in Fig 1a would be helpful for the reader). Is it just chance that both DBs presented here exhibit dynamics on two very disparate time scales? If slow and fast dynamics are characteristic, perhaps this indicates dopant interaction effects (e.g., hopping conduction)? The authors should make more clear how many DBs have been studied using this technique, and what range of dynamical time scales have been observed. A priori, I'd expect that some DBs would just show bi- or multi-exponential decays on the microsecond timescale, rather than dynamics on time scales varying by several orders of magnitude. Also, it seems to me that there should be a distinct

voltage threshold for each dopant, but the manuscript text (e.g., line 137, 141) suggests that there is only one. This threshold is determined by the DB-dopant separation, so there should be two distinct voltage thresholds if the dynamics of two dopants are responsible for the observations.

4) The authors don't discuss much why the different types of time-resolved measurements probe the filling/emptying rates of the dopant(s). Empirically, I can see that in Fig 3a the current is dropping from 40pA (high conductance) to 15pA (low conductance), but how does this connect with Fig. 4 showing a filling of the As dopant state from the CB for the corresponding rate? The manuscript has one line (#90) baldly stating that the DB is excited to a high conductance state by the pump. Why is this so? Given that this is the key novelty of this paper, I think more explicit discussion illustrating how the technique works is important. It would help if the authors included a much more detailed schematic than Fig. 1a and Fig 4, showing two dopants and the relevant rates that are probed in the experiments, as well as the prevailing TIBB and the band bending produced by the depletion region, including DB charge transition states, conduction and valence bands etc. Perhaps a simplified version would be appropriate for the main text, but a more complete version in the supplementary information? There's a lot going on here!

In terms of organization, the manuscript is difficult to understand unless one reads the supplementary information very carefully, as well as the groups' prior publications (particularly the 2015 New J Phys article). There is not enough discussion of the measurement details in either the methods section or the supp info. It took me some time for example, to realize how they were measuring probe current, and why that number was the same as what they measured in the conventional $I(t)$ measurement in Fig. 1. I actually like how the authors normalized for the duty cycle and duration of the probe pulse and subtracted the contribution from the pump, but this shouldn't be relegated to the supplementary information. I also found Figure 2 to be somewhat abstruse. For example, it is unclear if the data presented in Figure 2 really add anything to the discussion, and it's misleading to label this a pump/probe study, as the microsec-scale pulses are only separated by 10ns, which is less than the 25ns edgewidth the authors used. The labeling makes it a bit more difficult to understand what is being swept: the x-axis in Fig. 2c would be better as $V_{DC}+V_{probe}$, or V_{tot} . The effect being seen here could simply be due to the fact that the time-averaged bias voltage is different with the pulse train on or off: the pulses essentially are contiguous, and the duty cycle is $2/40 \sim 5\%$. A better comparison would be the shift in voltage onset for a conventional $I(V)$ measurement and one with the voltage pulses, but I don't really think this figure is necessary.

The manuscript by Rashidi et al., reports the time-resolved STM studies of individual dangling bonds in a Silicon surface. This is the first study of dopant dynamics in a semiconductor using a novel time resolved technique pioneered by co-author Loth for studies of magnetic adatoms. In my opinion, I believe the extension of this technique to study dynamics of individual dopants in a semiconductor is an important result that would be of broad interest to researchers in the nanoscience and semiconductor research communities.

We are grateful that the referee agrees the work provides important results and would be of broad interest.

However, I don't think the manuscript is clear enough to make the physics understandable to non-experts, and the omission of measurement details and organization would make it very difficult to understand in a reasonable amount of time for experts in the field. In my opinion the manuscript needs some structural reorganization and more significant discussion of points I list below to possibly be suitable for publication in an NPG journal.

We thank the referee the clear statement of the problem areas with the manuscript and the detailed suggestions for improvements. We have taken the advantage of the longer format of Nature Communications to reconstruct the manuscript and added more details to address each point raised by the referee. We believe this makes the paper more readily accessible to a wider research community. In particular, we moved measurement details from the supplementary information to the main text. We also add a full discussion section for the DC measurements in the beginning of the manuscript to cover the needed background information. This requires a concomitant introduction to the model we used added to the beginning of manuscript. This allows a full introduction of the many details of band bending, dopants etc. at play in this experiment, which is useful for both general and expert readers.

1) I think the authors could make more explicit references to their earlier work to make it clear that they have some idea of the concentration of the dopants in the depletion region (i.e. include a number even if only an order of magnitude), and I would feature the calculations in Fig. S7 more prominently in the main text. Given their concentration estimates and this calculation in fact, one can estimate that it's reasonable for only a handful of As dopants to influence the DB tunneling, and that the subsurface As which they can image with STM wouldn't be the primary contributors. According to this estimate, it would be dopants ~ 30nm directly below the tip which would have the biggest effect; if this is true, I'd revise line 148 which estimates 'several nm'.

Figure S7 is modified (dopant is located at a depth of 15nm below the surface) and added to the revised main manuscript as Fig. 2e. The following sections are revised to address the referee's comment.

“Secondary ion mass spectroscopy (SIMS) shows that as a result of flashing to 1250°C the sample is depleted of dopants in the near surface region (up to depth of ~60

nm)²¹. The black curve in Fig. 2e shows the conduction band (CB) edge calculated near the surface considering the band gap narrowing²², tip induced band bending (TIBB)²³ and the dopant concentration profile from SIMS measurements^{20,21}. As a result of the dopant depletion region the degenerately doped bulk does not extend to the surface. The conduction band edge first rises as a result of the depletion of dopants in that region, and falls rapidly as a result of TIBB at the surface, acting as an energy barrier for the electrons to conduct from the bulk (degenerately doped region) to the surface. This potential barrier accounts for the weak supply of electrons from the bulk to the DB for sample biases less negative than the critical voltage. As the critical voltage is crossed, there is a sudden change in the supply rate of the DB. Similar features in other systems have been attributed to the ionization of dopants in the field of the tip³⁻⁶. The electrostatic potential due to an ionized dopant significantly reduces this energy barrier (green curve in Fig. 2e), leading to an increase in the electron conductivity from the bulk to the surface. This understanding of the sharp onset in $I(V)$ measurements also provides a potential explanation for the related current fluctuations shown in Fig. 2c. Tunnelling current fluctuates, exhibiting two state noise, suggesting it could be due to an individual dopant switching charge state between neutral and positive. The magnitude of the influence of a nearby dopant ionization can be estimated by computing transmission through the energy barrier, using the WKB approximation (Supplementary Fig. 1). Additionally, the population of thermally excited electrons in the conduction band edge rises significantly for barriers of less than 5 meV at our experimental temperature (4.5 K). In conjunction with ionizing dopants, either tunnelling or thermal excitation can explain the electron transport from the bulk to the surface through the energy barrier and the observed sudden current turn on in $I(V)$ spectroscopy.

Our model of dopant ionization allows us to investigate the nature of the dopants causing the gating. Subsurface dopants within the first few atomic layers which can be imaged using STM^{5,6,26} are already ionized because of strong TIBB in our experiment. Moreover, calculations using a WKB approximation to compute transmission from the bulk to the DB at the surface (Supplementary Fig. 1) show that near surface dopants have a small gating effect. We therefore cannot correlate the gating effect with any nearby shallow subsurface dopants that can be imaged with standard STM techniques. We estimate that at the critical bias voltage only the dopants within approximately 15 nm of the surface can be ionized by the tip field. Therefore, only dopants that are located between 5 and 15 nm below the surface within a 15 nm lateral region are estimated to play a major role in opening the conduction channel. Based on the dopant profile measured by SIMS^{20,21}, there exist on average less than 10 dopants in this volume. Since these dopants are randomly distributed a large variety of switching behaviours are expected and indeed observed for different DBs.”

2) The text seems to be a bit unclear on whether there are two or three DB charge states (e.g., +1,

0, -1) accessible in the STM experiments. This is left ambiguous in line 63 in the text, but seems to be better discussed in the groups' recent PRL and New J Phys publications. A much more complete schematic of the expected tunneling channels and relevant rates would be helpful in this manuscript, more along the lines of what the group has done in their recent publications. This would help the expert appreciate why the switching behavior reported here may be qualitatively different than very similar behaviors observed under different tunneling conditions in the earlier work.

New Fig. 1 and Fig. 2(e,f) schematically show the expected tunnelling channels and rates. In addition to the new figures, we add the following sentences to the manuscript to discuss about the DB charge transition levels as well as to clarify why the switching behaviour reported here is qualitatively different than our earlier work:

“Alternative explanations of the sharp step in $I(V)$ spectra and of the current fluctuations involving transitions of the charge state of the DB itself need to be examined. The DB has three allowed charge states, positive, neutral, and negative, associated with two charge transition levels, (+/0) and (0/-). A sharp step in $I(V)$ can often be attributed to the tip Fermi level coming into resonance with a charge transition level²⁴. However, we are able to rule out such an explanation here because the tip Fermi level at the critical bias regime is already significantly below both charge transition levels of the DB²⁰ (Fig. 2f).

DB charge state fluctuation itself can also be ruled out as a possible source of the slow dynamics (Fig. 2c and 2d). The current time traces that show the telegraph noise are recorded with the tip placed exactly on top of the DB, with the initial tip height set at a voltage where the DB shows a bright appearance. Therefore, the tip is removed from the surface by an additional ~ 200 pm compared to the hydrogen-terminated silicon surface. This effectively eliminates electrons tunnelling directly from the silicon valence band. The only sufficiently conductive tunnelling pathway that remains is the one that originates from the conduction band and passes through the DB. Since the tunnel current passes through the DB, the fluctuations in the DB charge state must occur at timescales less than $\epsilon/IT \approx 2$ ns to 20 ns; much faster than the switching observed. This contrasts with a recent STM study²⁵ that considered millisecond dynamics of the DB caused by changes in its charge state. Crucially, to detect this effect the STM tip had to be placed 2 to 4 nm away from the DB, laterally. As a result, the current which flowed through the DB was negligible ($\sim 10^{-6}$ pA to 10^{-4} pA, far below our sensitivity), and the much larger direct tip-sample tunnelling current was gated by the electrostatic potential of the charged DB. In this work, since the STM tip is placed at the DB, and for the reasons outlined above, we can rule out this type of explanation. [...]

3) A key point that lacks discussion in the manuscript is some rationale for why they see two such disparate timescales for dynamics. The authors suggest this is due to the influence of two dopants (inclusion of two As dopants in Fig 1a would be helpful for the reader).

We included two dopants Fig. 1 in the revised manuscript with different filling rate (Γ_{HL}) from the bulk.

Is it just chance that both DBs presented here exhibit dynamics on two very disparate time scales?

The example showed in the supplementary information exhibits only slow dynamics. We were not able to measure any fast dynamics on this DB within our measurement sensitivity. The DB shown in the main manuscript is indeed fortuitously exhibiting dynamics on two distinct time scales, which we attribute it to the influence of two isolated dopants with different supply rate.

If slow and fast dynamics are characteristic, perhaps this indicates dopant interaction effects (e.g., hopping conduction)? The authors should make more clear how many DBs have been studied using this technique, and what range of dynamical time scales have been observed.

We have inspected hundreds of DBs with this technique. Indeed, we do see a variety of timescales and thresholds. The two selected examples for this work to some extent cover the wide range of threshold voltages and timescales we have observed.

The dynamics are characteristic of the system but slow and fast dynamics are not always present on a particular DB. The majority of DBs that we have studied exhibit only slow dynamics but we have observed cases with only fast dynamics as well.

The following paragraph in the revised manuscript is revised to address this:

“Dopant ionization is exponentially dependent on the applied bias (Fig. 5e and Equation 1), but dopant supply is determined by the local environment and is roughly independent of bias (Fig. 2g) as the tip field extends only approximately 15 nm below the surface. Therefore, the onset of current occurs in a narrow transition region near the critical voltage where ionization overtakes dopant supply. We have examined a few hundred DBs by our technique. In most cases, the supply rate of electrons to the dopant was in the Hz to kHz range, and we were able to observe the corresponding dynamics using standard $I(t)$ measurements. In few cases, the supply rate was in the kHz to MHz range, and we have used pump probe to observe these dynamics near the cross-over of these two rates. Variation in DB-dopant separation leads to variation in the opening voltage (observed between -0.7 V and -2.0 V) of this channel in standard $I(V)$ STS measurements among individual DBs²⁰.”

A priori, I'd expect that some DBs would just show bi- or multi-exponential decays on the microsecond timescale, rather than dynamics on time scales varying by several orders of magnitude.”

Due to the relative weakness of the faster dynamics found, actually resolving bi/multi exponential decays is not really practical and showing that a single exponential is not a good fit is not easy. In principle, they should be visible, but this requires two dopants to have similar

contributions, similar voltage thresholds, and similar timescales. This makes it difficult to find examples.

Also, it seems to me that there should be a distinct voltage threshold for each dopant, but the manuscript text (e.g., line 137, 141) suggests that there is only one. This threshold is determined by the DB-dopant separation, so there should be two distinct voltage thresholds if the dynamics of two dopants are responsible for the observations.

We thank the reviewer for bringing to our attention the lack of enough discussion about this point. We add the following paragraph to the revised manuscript to discuss this:

The DB shown in Fig. 2 exhibits dynamics at two distinct time scales. We attribute this to the influence of two isolated dopants with different supply rates one slow, and one fast from the bulk. These two dopants are schematically displayed in Fig. 1. Our results show that there is a distinct voltage threshold for each dopant. Figure 5d and 5e show that we begin to find fast dynamics at ~ -2.05 V, *i.e.*, pumping to less negative values than this threshold voltage does not result in any measurable dopant ionization time constant. This is notable in Fig. 4b also as the splitting between the curves remaining open for bias voltages more negative than ~ -2.05 V where slow telegraph noise is no longer measurable (Fig. 2d and 2g). Therefore, we assign ~ -1.80 V where the slow dynamics start to appear (Fig. 2d) as the threshold voltage to ionize the slow dopant and ~ -2.05 V to the ionization of the fast dopant.

4) The authors don't discuss much why the different types of time-resolved measurements probe the filling/emptying rates of the dopant(s). Empirically, I can see that in Fig 3a the current is dropping from 40pA (high conductance) to 15pA (low conductance), but how does this connect with Fig. 4 showing a filling of the As dopant state from the CB for the corresponding rate? The manuscript has one line (#90) baldly stating that the DB is excited to a high conductance state by the pump. Why is this so? Given that this is the key novelty of this paper, I think more explicit discussion illustrating how the technique works is important.

We add the following paragraph to the revised manuscript to explicitly discuss how the technique works. We also moved a full discussion of the pump-probe technique to the revised manuscript from the supplementary information.

“We apply three variations of pump-probe STM: (i) a pulsed form of $I(V)$ spectroscopy used to search efficiently for fast dynamics, (ii) a typical pump-probe experiment to measure the relaxation dynamics of the high conductivity state, and (iii) a variable pump width experiment to measure the excitation dynamics of the transition from the low conductance state to the high conductance state. The experimental scheme for applying pulses to the STM is very simple. Pulses are applied directly to the STM tip, and a pair of radio frequency (RF) switches are used to turn off and on the pump and

probe pulse trains (Fig. 3). Common to all three types of experiment is the role of the pump and probe. The pump transiently brings the bias level from below to above the critical voltage, driving the transition to the high conductance state. In the context of a dopant gate, the pump increases TIBB making it more likely for a dopant to ionize and improve electron supply to the DB. The probe follows at a lower voltage to sample the conductance after the pump.”

It would help if the authors included a much more detailed schematic than Fig. 1a and Fig 4, showing two dopants and the relevant rates that are probed in the experiments, as well as the prevailing TIBB and the band bending produced by the depletion region, including DB charge transition states, conduction and valence bands etc. Perhaps a simplified version would be appropriate for the main text, but a more complete version in the supplementary information? There's a lot going on here!

Previous Fig. 1a and Fig. 4 are replaced by Fig. 1 and Fig. 2(e,f) in the revised manuscript to schematically display all the suggested points. In particular, Fig. 1 displays two dopants and the relevant rates probed in the experiments. Figure 2(e,f) indicates the TIBB, band bending produced by dopant depletion region, DB charge transition levels, as well as conduction and valence bands.

In terms of organization, the manuscript is difficult to understand unless one reads the supplementary information very carefully, as well as the groups' prior publications (particularly the 2015 New J Phys article).

We have moved the important parts from the supplementary information to the main text. We also add a full discussion section for the DC measurements in the beginning of the manuscript to cover the background information that the readers need to better understand the paper. We also shift the model section to the beginning of manuscript. This helps the readers to understand the time resolved results better.

There is not enough discussion of the measurement details in either the methods section or the supp info. It took me some time for example, to realize how they were measuring probe current, and why that number was the same as what they measured in the conventional $I(t)$ measurement in Fig. 1. I actually like how the authors normalized for the duty cycle and duration of the probe pulse and subtracted the contribution from the pump, but this shouldn't be relegated to the supplementary information.

The reviewer is correct that due to brevity we did not have enough discussion of the measurement details in the main manuscript. We are pleased that he/she likes the normalization method we have used to directly match fast measurements with DC. We modified and moved all

the measurements details, including the normalization method from the supplementary information to the main text:

“In order to obtain the actual current from the probe pulse, the measured current from the preamplifier is divided by the duty cycle of the pulse series and the current from the pump pulse is subtracted from that. The time-resolved $I(V)$ and the current obtained by DC $I(V)$ at bias voltages where no dynamics are expected, closely match, verifying this normalization scheme.”

I also found Figure 2 to be somewhat abstruse. For example, it is unclear if the data presented in Figure 2 really add anything to the discussion, and it's misleading to label this a pump/probe study, as the microsec-scale pulses are only separated by 10ns, which is less than the 25ns edge width the authors used. The labeling makes it a bit more difficult to understand what is being swept: the x-axis in Fig. 2c would be better as $V_{DC}+V_{probe}$, or V_{tot} . The effect being seen here could simply be due to the fact that the time-averaged bias voltage is different with the pulse train on or off: the pulses essentially are contiguous, and the duty cycle is $2/40 \sim 5\%$. A better comparison would be the shift in voltage onset for a conventional $I(V)$ measurement and one with the voltage pulses, but I don't really think this figure is necessary.

In this case we would like to point out that the 10 ns delay is measured between the absolute limits of the pulses, including the tapered edges, i.e., 10 ns after there is no pulse overlap. Also, the current from the pump pulse is subtracted from the signal. The close match between the two curves outside the critical bias voltage regime rules out any kind of artefact. To better show this we have extended the x-axis in the revised manuscript. The kink at ~ -2.25 V is present on both curves, showing that the effective voltage for the pump+probe curve is not different than the probe only curve. Figure 5b in the revised manuscript helps to better understand this figure. It shows that the hysteresis only happens for short relative delays between pump and probe. For longer relative delays the hysteresis is extinguished.

We thank the reviewer for pointing out the confusion about labelling. We revise the figure with the new labelling suggested by the reviewer.

With regard to the practicality, we argue that we use the time-resolved $I(V)$ spectroscopy as a search tool to find the right DBs with time dynamics falling in our measurement bandwidth. Also, to the best of our knowledge this is the first time-resolved $I(V)$ STM tunnelling spectroscopy. It was sufficiently useful, as the dynamics are not straightforward to find, that we feel it deserves a place in the manuscript on that merit alone and its potential utility in future studies.

We add the following sentences in the revised manuscript to clarify some of the abovementioned points:

“Figure 4a displays a schematic of time-resolved $I(V)$ spectroscopy used to search for dynamics. Instead of sweeping the DC bias voltage as in standard $I(V)$ spectroscopy, we keep the DC bias fixed and sweep the amplitude of a train of fast microsecond scale

voltage pulses. Preceding each probe pulse, a large fixed amplitude pump pulse transiently disturbs the system. The delay between pump and probe is fixed such that the pulses do not overlap (10 ns between pulse edges). Thus the transient state immediately resulting from the pump is interrogated by the probe. Since the current is almost zero at the DC offset, the measured tunnelling current from the preamplifier comes exclusively from current during the time when the pulses are on. Comparison to an equivalent curve measured with no pump pulse reveals any triggered dynamics.”